# Crystalline–Amorphous Nanostructures: Microstructure, Property and Modelling

**DOI:** 10.3390/ma16072874

**Published:** 2023-04-04

**Authors:** Bingqiang Wei, Lin Li, Lin Shao, Jian Wang

**Affiliations:** 1Department of Mechanical and Materials Engineering, University of Nebraska-Lincoln, Lincoln, NE 68588, USA; bwei5@huskers.unl.edu; 2School for Engineering of Matter, Transport and Energy, Arizona State University, Tempe, AZ 85287, USA; lin.li.10@asu.edu; 3Department of Nuclear Engineering, Texas A&M University, College Station, TX 77843, USA; lshao@tamu.edu

**Keywords:** crystalline–amorphous nanocomposites, microstructure, properties, modelling

## Abstract

Crystalline metals generally exhibit good deformability but low strength and poor irradiation tolerance. Amorphous materials in general display poor deformability but high strength and good irradiation tolerance. Interestingly, refining characteristic size can enhance the flow strength of crystalline metals and the deformability of amorphous materials. Thus, crystalline–amorphous nanostructures can exhibit an enhanced strength and an improved plastic flow stability. In addition, high-density interfaces can trap radiation-induced defects and accommodate free volume fluctuation. In this article, we review crystalline–amorphous nanocomposites with characteristic microstructures including nanolaminates, core–shell microstructures, and crystalline/amorphous-based dual-phase nanocomposites. The focus is put on synthesis of characteristic microstructures, deformation behaviors, and multiscale materials modelling.

## 1. Introduction

Strong, ductile, thermally stable, and irradiation tolerant materials are in urgent demand for improving the safety and efficiency of advanced nuclear reactors [1,2,3]. Materials that employ microstructure features to manage radiation damage and maintain high-temperature mechanical properties are especially desirable [3]. For crystalline materials, high strength is usually achieved by refining crystal size or introducing dispersed reinforcement precipitates/particles, corresponding to grain boundary/interface strengthening or second phase strengthening [4,5,6,7]. However, microstructural stability and plastic deformability of materials with high density interfaces are facing great challenges. Mechanically and/or thermally driven grain coarsening usually occurs due to the excess grain boundary (GB) free energy, resulting in softening behavior [8,9]. Due to the fine grain size, dislocations are most likely to nucleate at GBs, propagate across grains, and then be absorbed at the opposite GBs [10]. Dislocation accumulation at GBs and limited slip transmission across GBs results in strain localization at GBs and then promotes plastic flow instability along GBs and even facilitates crack initiation at GBs, leading to premature failure [11,12]. Thus, there are extensive efforts to improve mechanical properties of materials by tailoring structures and properties of GBs [13,14,15]. As for second phase strengthened metallic materials, intermetallic or covalent crystalline precipitates are intrinsically brittle at room temperature (RT). The significant difference in elasticity and plasticity between precipitates and matrix results in deformation incompatibility which leads to precipitate cracking or interface de-cohesion [16,17,18]. Thus, there is an urgent demand to improve co-deformability between matrix and second phase, such as reducing dimension of second phase [19,20,21]. In addition to mechanical properties, crystalline materials have issues due to incomplete defect self-repairing of interstitials and vacancies under irradiation, which leads to embrittlement and swelling from defect clustering [22,23]. Experimental and modeling studies demonstrated that interfaces including interphase boundaries, twin boundaries and grain boundaries can play roles in absorbing, trapping, and promoting reassembly of radiation-generated defects [24,25].

In contrast, amorphous materials generally exhibit superior resistance to wear and corrosion, and unique mechanical and chemical properties due to the absence of GBs and crystalline structures [26,27,28,29]. Especially high-crystallization-temperature amorphous materials could be very promising candidates for high radiation tolerance at high temperatures since they do not contain conventional crystal defects such as vacancies, interstitials, or dislocations which evolve in crystalline materials under irradiation [28,30,31,32]. With respect to the bonding mechanisms, amorphous materials can be classified into metallic glass, oxide ceramics and non-oxide ceramics. Metallic glasses (MGs) are composed of multiple principle metal elements that form metallic bonding. Recrystallization of a glassy phase occurs at annealing or elevated temperature. Under ion or neutron irradiation, thermal spikes form due to damage cascade formation. However, whether localized recrystallization occurs or not is complicated and controlled by the cooling rate of damage cascade region [33,34]. This complexity leads to unique radiation tolerance of MGs. Amorphous ceramic is an inorganic, non-metallic, solid material comprising non-metal or metalloid atoms primarily held in ionic and covalent bonds. Non-metallic-bonded structures tend to have rather high melting points and high strength but poor deformability and low fracture toughness [35,36,37]. Plastic deformation in amorphous materials is generally accommodated discretely by the so-called shear transformation zones (STZ) and/or localized shear banding (flow instability) [38,39,40,41]. The STZ mechanism is connected to microstructural heterogeneities. The formation and propagation of shear bands is ascribed to statistically occurring, spatially homogeneous nucleation of STZs and their coalescence [41,42,43,44]. Experimental and modelling studies revealed that amorphous materials exhibit high strength without strain hardening capability and “brittle-like” behavior [45,46].

Improving the strength and plasticity of crystalline and amorphous materials can be realized through composition engineering and microstructural engineering [47,48,49,50,51]. In contrast, microstructural engineering shows more promise, such as NC materials with thermally stabilized amorphous GBs or crystal–amorphous nanolaminates [52,53,54,55]. Compared to ordered crystal–crystal interface, amorphous boundary layers can effectively reduce interface energy, block the motion of dislocations, trap/absorb and smear accumulated dislocations away along crystal–amorphous interfaces (CAIs) [56,57]. More importantly, refined/constrained nanosized amorphous phase shows plastic flow ability [58,59]. In contrast to the brittle intermetallic compound, the amorphous nanoparticles as second phase can co-deform with metallic matrix, enhancing strength while maintaining large deformability [60]. In addition, the ductile nanosized crystal or dendrites in amorphous matrix can inhibit nucleation/propagation of shear bands and co-deformed with amorphous matrix, improving plasticity of amorphous materials [47,61,62,63]. Given the superior radiation tolerance exhibited in amorphous ceramic alloys, crystalline metal/amorphous ceramic composites have potential applications in extreme irradiation environments due to better heat conductivity and improved thermo-mechanical properties [64,65,66]. CAIs can act as defect sink because amorphous phase can trap and relax away conventional crystal defects (such as vacancies and interstitials) that are generated in crystalline phase. In the Fe-SiOC nanolaminates [65,67] and Ni-SiOC nanocomposites [68], the CAIs act as stable and efficient sinks for defects, mitigating irradiation damage and suppressing void swelling during irradiation. In addition, the interface could suppress crystallization of amorphous layer within the interface-affected zones, as observed in Fe/Y_2_O_3_ nanolayers with individual layer thicknesses of 10 and 50 nm [69], because interfaces absorb radiation-induced defects and produce interfacial stress to mitigate radiation-induced crystallization.

Based on characteristic microstructures, the reported crystalline–amorphous dual-phase nanocomposite system includes crystalline–amorphous nanolaminates [55,56,70,71], crystal–amorphous core–shell structures [13,14,53,72] and crystal- or amorphous-based dual-phase nanocomposites [60,62,63,73]. In this article, we briefly review crystalline–amorphous dual-phase nanocomposites with the three typical microstructures. The focus is put on synthesis of characteristic microstructures, deformation behaviors, and multiscale materials modelling.

## 2. Characteristic Microstructures

### 2.1. Nanolaminates

Crystalline–amorphous nanolaminates are composed of alternatively crystalline and amorphous nanolayers. Compared to a liquid-to-solid melting/casting routine, physical vapor deposition (PVD) such as magnetron sputtering allows for the deposition of crystalline–amorphous layer with a wide range of composition, thickness and volume fraction [74,75]. A typical sputtering process entails the condensation of gaseous particles into a solid state at a cooling rate as fast as ~10^10^–10^12^ K/s. The control of thickness of the crystalline–amorphous nanolayer is usually realized by changing the input power, deposition time and substrate–target distance. The deposition selection of crystalline vs. amorphous layer is controlled by switching shutter of individual evaporation cells. Amorphous layers including metallic glasses (MGs) [56,70,74], amorphous ceramics [66,69,76], and carbon or silicon family elemental glasses [71,77,78] have been successfully fabricated in multi-layered crystalline–amorphous composites, as shown in Figure 1. Due to the non-equilibrium fabrication process, crystalline–amorphous nanolaminates may exhibit poor thermal stability under heating, which can be attributed to grain growth/coarsening of NC metals in the crystal nanolayer [79], crystallization of amorphous layers [80], and chemical reaction between crystalline and amorphous layers [81]. Hence, the composition selection and layer morphology control need to be optimized.

### 2.2. Core–Shell Nanostructures

Dual-phase core–shell nanostructures are usually composed of crystalline phase as the core surrounded by amorphous phase as the shell. The shell is generally a few nanometers thick. Such characteristic microstructures can be synthesized through GB complexion [52,82] or spontaneous phase separation [14,53]. GB segregation by dopant elements during high temperature annealing can promote formation of amorphous intergranular thin film (AIF), i.e., amorphous GB complexion, as demonstrated in a variety of binary and ternary metallic alloys such as Ni-W, Ni-Mo, Ni-Zr, Cu-Zr, Cu-Zr-Hf and Al-(Mg, Fe, Ni)-Y [83,84,85,86,87,88]. The formation of AIF is compositionally selective under the thermodynamic and kinetic constraints, i.e., encouraging the segregation of dopants to interfaces and lowering the formation energy for a glassy structure [82,89,90,91]. Correspondingly, Schuler and Rupert [85] proposed that dopant elements with a positive enthalpy of segregation and negative enthalpy of mixing promote the formation of amorphous GB complexions. The thickness and stability of AIF are mainly dependent on chemical complexity at GB, i.e., increasing the chemical complexity can increase the thickness and stability of AIF. For example, the AIF thickness in Cu-Zr-Hf ternary alloy is larger than that in Cu-Zr binary alloy (Figure 2) [86,92]. It is worth mentioning that the crystalline–amorphous nanocomposites synthesized by amorphous GB complexion generally exhibit high thermal stability [93,94], because amorphous GB complexion can extensively inhibit grain growth of NC metals by reducing GB energy and kinetically slowing GB migration based on solute drag. More importantly, the amorphous GB complexion is usually formed under high-temperature annealing.

Sputtering deposition is another method to synthesize core–shell nanostructures, corresponding to a self-assembly process associated with phase separation [14,53,72,95]. When Al-based alloy target with higher Al content (e.g., 95 at.%) is used for sputtering deposition, Al nanocrystals nucleate in the matrix of Al-based MG, forming a crystalline–amorphous core–shell nanostructure [14]. Since MGs usually show low thermal stability, fast crystallization will occur in MGs shell under heating, leading to low thermal stability of crystal–MGs nanostructures [14,96]. In contrast, amorphous ceramic has much higher thermal stability and can form along GBs of nanocrystalline metals during sputtering [53,72]. Recently, amorphous SiOC ceramic with a wide range of compositions was synthesized by co-sputtering SiC and SiO_2_ targets and exhibit superb thermal stability with crystallization temperature above 1300 °C [28]. When Si-O-C were co-sputtered with metal elements (Fe or Ni), the bonding energy difference between metallic and non-metallic elements promotes phase separation [53,72,97]. With increasing the content of metallic element up to 70–80 at.%, the core–shell nanostructures form (Figure 3) [53,72]. The as-deposited Fe(80 at.%)-SiOC film is composed of crystalline Fe nanocolumns and nanoscale shell of amorphous ceramic SiOC, exhibiting a core–shell nanocolumnar structure [72], while the as-deposited Ni(75 at.%)-SiOC film comprises crystalline Ni nanocrystals and nanoscale grain boundaries of amorphous ceramic SiOC, exhibiting a core–shell nanocrystalline structure [53]. High-temperature annealing experiments demonstrated that the Fe(80 at.%)-SiOC and Ni(75 at.%)-SiOC core–shell nanostructures display excellent thermo-mechanical stability up to annealing temperature of 600 °C [60,72]. The high thermal stability is ascribed to the high crystallization temperature of amorphous SiOC and its kinetically pinning effect on inhibiting grain growth of NC metals.

It is worth mentioning that cellular structure associated with chemical segregation at cellular boundary was widely observed in laser-assisted manufacturing and processing processes [98,99,100]. With the “right” chemical segregation, the amorphous shell can form [100,101]. As shown in Figure 4a,b, Sun et al. [101] fabricated a heterogeneous gradient nanostructured layer (600 μm-thick) by laser surface remelting an austenitic Hadfield manganese steel. The nanostructured layer exhibits a nanocrystalline–amorphous core–shell microstructure. In addition, a core–shell microstructure shown in Figure 4c,d is widely observed in austenitic 316 L stainless steels fabricated by a laser powder-bed-fusion (L-PBF) technique [100]. Although the shell is not an amorphous phase, the composition is generally different from the core due to segregation of alloying elements, such as segregation of Mo and Cr to the solidification cellular walls and formation of amorphous nanoparticles along low-angle grain boundaries. This could provide a strategy to fabricate bulk core–shell crystal–amorphous nanocomposites by advanced laser-assisted manufacturing technique with high efficiency, low cost, and high flexibility.

### 2.3. Dual-Phase Nanocomposites

Analogous to metallic alloy with precipitates, crystalline or amorphous nanoparticles can form in amorphous or crystalline matrix, producing crystalline–amorphous dual-phase nanocomposites [60,62,63,73]. Since the non-equilibrium fabrication process of MGs requires fast cooling during rapid solidification or sputtering deposition, nanocrystalline phase can form and grow in MG matrix via manipulating cooling rate, annealing temperature/time and adjusting composition [62,103,104,105]. Ming et al. [62] synthesized TiZr-based crystalline–amorphous composites by melt spinning with a mediate cooling rate of ~10^6^ K/s, which consists of micro-sized equiaxed grains and nano-width amorphous grain boundary. Notably, each grain displays crystalline–amorphous nanostructure comprising nanosized metastable crystalline (~36 nm) and amorphous (~25 nm) phases that are arranged in the form of 3D-interconnected nano-bands (Figure 5). In situ micromechanical tests (tension and compression) of the TiZr-based dual-phase composites revealed the enhanced ductility and strain hardening capability than both amorphous and crystalline phases. The crystalline–amorphous nanostructure exhibits an ultra-high yield strength of ~1.80 GPa, an ultimate tensile strength of ~2.3 GPa, and a large uniform strain of ~7.0%. By finite element analysis, the authors claimed the synergetic deformation mechanisms between amorphous phase and crystalline phase. The amorphous phase constrains crystalline domains, imposing extra strain-hardening to crystalline domains, while crystalline domains plastically deform on certain slip planes, preventing the premature shear localization in the amorphous phases.

Amorphous-to-crystalline transition occurs when highly correlated atomic movements are allowed. Heating is the most obvious example of crystallization. Correspondingly, annealing the amorphous precursor is another approach to form nanocrystals in amorphous matrix based on partial crystallization [73,103]. Notably, the size and volume fraction of nanocrystals can be modified by adjusting the annealing temperature and/or time. Li et al. fabricated a series of Fe-based crystal–amorphous nanocomposites through nanocrystallization in Fe-based amorphous ribbons via annealing [103]. The diameter and volume fraction of Fe(Si) nanocrystals increase with increasing annealing time, for example, amorphous feature size is reduced from 10 nm to less than 0.5 nm as the volume fraction of nanocrystals increases from 16% to 95%. Xu et al. [106] reported that bulk amorphous Al_2_O_3_-ZrO_2_-Y_2_O_3_ remain a fully amorphous structure when hot pressed at 350 °C, while change to amorphous matrix composite with nanocrystallite when hot pressed was conducted at 400 °C or 450 °C. In addition to annealing, amorphous phase may experience irradiation-induced amorphous-to-crystalline transition under high energy particle irradiation [107,108,109]. Nanocrystals were observed in Ni_52.5_Nb_10_Zr_15_Ti_15_Pt_7.5_ amorphous alloy (Figure 6a) after the alloy was subject to 1 MeV Ni ion irradiation [108]. Three distinctive regions are marked in the irradiated specimen (Figure 6b), including amorphous matrix with bright contrast (Region 1), irregular precipitates with relatively darker contrast (Region 2), and nanocrystals (Region 3) within region 2, as evidenced from dark field TEM image in Figure 6c. Composition analysis shows that regions 2 and 3 share the same composition, but they are different from region 1. Correspondingly, it is speculated that irradiation induces formation of nanocluster, then nanocrystallization occurs within these nanoclusters. Alternatively, it is possible that nanocrystals form first, diffuse and then agglomerate into region 2, followed by re-arrangement of nanocrystals and crystal growth in region 3. The two-stage growth is further supported by a two-step irradiation test of Cu_50_Zr_45_Ti_5_ metallic glass, precipitation in the first irradiation step and nanocrystallization within the precipitates in second irradiation step [109]. Figure 6d shows formation of darker precipitates of typical size of 10 nm after 2 keV Ar ion irradiation [109]. The corresponding diffraction pattern collected from the dark contrast region shows these precipitates still maintain glassy states. In the second-step irradiation, the Ar-irradiated sample is re-irradiated by 200 keV electron beams in a TEM chamber. Figure 6e shows that the electron irradiation induces formation of even smaller nanometer features within the large precipitates. A few such nanometer features are marked by arrows. The diffraction pattern suggests nanocrystals begin to form. As marked by arrows in the dark-field TEM micrograph in Figure 6f, the nanometer size features appear as white dots, suggesting their nanocrystal nature.

Sputtering deposition also provides a convenient route for in situ formation of dual-phase nanocomposites by regulating composition. With increasing the atomic fraction of Mo in Al-Mo binary system [110], microstructures of the as-deposited Al-Mo binary alloy change from polycrystalline Al to Al-rich crystalline matrix with local amorphous regions, and to a full amorphous alloy when the atomic fraction of Mo increases up to 32 at.%. Owing to the bonding energy difference between metallic and non-metallic elements (such as Si, O, C, and N), when metal elements (such as Cu, Fe, and Ni) are co-sputtered with Si-O-C elements, metallic atoms decorate amorphous SiOC ceramic associated with the formation of nanosized metal-rich clusters or metal nanocrystals in amorphous ceramic matrix, as shown in Figure 7, depending on the type of metal element and its atomic fraction [53,60,63,72,111]. The as-deposited Fe-SiOC films with the Fe concentrations of 4–34 at.% exhibit amorphous structure with spatially distributed Fe-rich amorphous nanoclusters, referred to as dual amorphous nanocomposites (Figure 7a,b). Further high temperature annealing can tune Fe-rich amorphous nanoclusters into Fe nanocrystals, forming crystalline–amorphous nanocomposites (Figure 7c,d). With increasing the content of Fe up to 70–80 at.%, the as-deposited Fe-SiOC film exhibits a core–shell nanocolumnar structure (Figure 3) comprising crystalline Fe nanocolumns surrounded with nanoscale amorphous ceramic SiOC [63,72]. Compared with Fe-SiOC, the as-deposited Cu-SiOC films show quite a different microstructure where large numbers of Cu nanocrystals are non-uniformly distributed in the amorphous ceramic matrix, referred to as crystalline–amorphous nanocomposites (Figure 7e) [111]. Compared with the formation of Fe-rich amorphous nanoclusters, the relatively low bond energy between Cu and non-metallic elements Si, O and C atoms (Cu-Si/Fe-Si: 224/310 kJ/mol; Cu-O/Fe-O: 287/407 kJ/mol, Cu-C/Fe-C: 45/390 kJ/mol) facilitates clustering Cu atoms and then nucleating Cu nanocrystals during deposition [97,112].

Analogous to nanocrystals embedded in amorphous matrix, amorphous nanoparticles can also be introduced in polycrystalline metals. Wei et al. [60] demonstrated that the as-deposited Ni(75 at.%)-SiOC sample with a core–shell microstructure can be further tuned to amorphous particles embedded in polycrystalline Ni by high-temperature annealing at 600 °C, 800 °C and 1000 °C under vacuum for 1 h, respectively (Figure 8). Corresponding to the high crystallization temperature of amorphous SiOC, SiOC maintains a stable amorphous structure, ensuring crystal–amorphous dual-phase structures while the core–shell microstructure is weakened and even diminished with increasing the annealing temperature. The core–shell nanostructure remains stable up to 400 °C. Although there is no obvious growth of Ni nanograins, grain boundaries of amorphous SiOC phase tend to be thinner while thicken at the triple junctions of Ni nanograins (Figure 8a). As annealing temperature increases to 600 °C, Ni crystals grow and amorphous SiOC nanoparticles form and grow up along GBs, especially at the triple junctions of Ni nanograins (Figure 8b). Moreover, amorphous SiOC phase was occasionally observed along GBs (Figure 8g). When further increasing the annealing temperature up to 800 °C and 1000 °C, both Ni crystals and SiOC amorphous particles grew up, especially Ni grains coarsen and engulf amorphous particles (Figure 8c,h). It is noted that there are no intermetallic or compounds in annealed samples as evidenced by the distribution of Ni atoms which are separated from Si, O and C amorphous elements. Figure 8d–f schematically illustrate microstructure evolution of Ni-SiOC nanocomposites, respectively. Correspondingly, the feature dimensions of crystalline Ni and amorphous SiOC grew up from 13 nm Ni nanograins and 2 nm thick SiOC boundary layers in the as-deposited and 400 °C annealed samples to ~22 nm Ni nanograins and ~9 nm SiOC nanoparticles in the 600 °C annealed sample, to ~220 nm Ni nanograins and ~110 nm SiOC nanoparticles in the 800 °C annealed sample and to ~500 nm Ni grains and ~200 nm SiOC particles in the 1000 °C annealed sample.

Note that amorphous nanoparticles can also be introduced into metallic matrix (Figure 4d) by laser-assisted manufacturing [100,102]. However, their density is relatively low and the feature size of metallic matrix is usually in micron scale.

## 3. Deformation Behaviors and Mechanisms

Plastic deformation is generally accommodated by dislocation motion in crystalline materials [113,114], while in amorphous materials it is accommodated by formation and propagation of STZs and shear banding [38,40,41]. Interfaces as a planar defect hold different atomic structures and thermodynamic properties from adjacent phases, and thus can act as sources associated with pre-existing interface defects for nucleating plastic deformation carriers (such as dislocations, twins, and phase transformation bands in crystalline phase or STZs and shear bands in amorphous phase) [5,115,116,117], strong barriers for blocking and impeding the motion of plastic deformation carriers [118,119,120,121,122], and sinks for absorbing, trapping and reassembly (that may lead to recovery) of defects [3,25,56,123]. For crystalline–amorphous nanocomposites, dislocations nucleated at CAIs will propagate across crystals, generating plastic deformation. Due to the lack of slip systems in amorphous phase, dislocations gliding in crystal phase will be blocked by and deposited at CAIs. Dislocations piled up at interfaces create stress/strain concentrations, facilitating nucleation of STZs and promoting localized shearing in amorphous phase. Since there is no strain hardening capability in amorphous phase, shear banding propagates quickly once it forms. In what follows, we review plastic deformation behaviors and mechanisms with respect to characteristic microstructure of dual-phase crystalline–amorphous composites.

### 3.1. Crystal–Amorphous Nanolaminate

Interface spacing or layer thickness is the key microstructural parameter in changing dominant deformation mechanisms of each constitutive layer and correspondingly determining mechanical properties of laminated composites [70,124]. For crystalline layers, plastic deformation is realized through dislocations that propagate in the confined layer. Consequently, interfaces acting as strong barriers block dislocation motion, resulting in dislocations accumulation at interfaces. Correspondingly, these accumulated dislocations generate local stress/strain concentrations, facilitating plastic deformation in amorphous layers. Take Cu-CuZr nanolaminate as an example [70]; decreasing the amorphous CuZr layer thickness leads to a transition from shear banding to homogenous plastic deformation. The lack of shear banding is ascribed to the intrinsic small size effect of amorphous phase and dislocation–interface interaction. When the feature size of amorphous material is refined into nanoscale, amorphous structure deforms via STZs which is activated by dislocation–interface interactions, because shear banding by coalescence of STZs will be inhibited due to the fine thickness [58,59]. The additional shear stress produced by plastic incompatibility between Cu and CuZr layers and deposited dislocations on CAIs facilitates STZs in thinner amorphous CuZr layer, promoting co-deformation between crystal and amorphous layers.

The detailed deformation mechanism can be rationalized with respect to layer thickness as follows. Disregarding layer thickness, dislocations first nucleate and propagate in crystalline layer (Cu layers) under applied loading and these dislocations propagate in crystalline layers and are deposited at CAIs (Cu-CuZr interfaces). Correspondingly, plastic deformation develops in crystalline layers (Cu layers) while amorphous layers still elastically deform. As consequence, plastic incompatibility between crystalline layers and amorphous layers is accommodated by elastic deformation, resulting in high stress in amorphous layers. In addition to the stress associated with plastic incompatibility, another stress contribution is associated with the deposited dislocation arrays at CAIs, and the stress magnitude associated with the dislocation arrays is proportional to the layer thickness ratio of *h*_Cu_/*h*_CuZr_ [125]. The higher stress (tension or compression is opposite to the applied loading) is thus generated for the thinner amorphous layer. More importantly, the deposited dislocations on the two adjacent CAIs interact and generate the shear stress on the plane non-parallel to amorphous layers. Putting all these stress contributions together, the shear stress on a plane non-parallel to the amorphous layer includes the stress associated with plastic incompatibility, the shear stress due to dislocation interaction, and the applied stress. Corresponding to the variation of stress field of a dislocation with the distance, the shear stress due to dislocation interaction increases with reducing amorphous layer thickness. When the layer is refined into a few nanometers, the shear stress becomes a significant contributor [126]. Thus, for the thinner amorphous layers, the larger shear stress on the plane non-parallel to the amorphous layers will facilitate the activation of STZs and promote the propagation of STZs through the layer, achieving plastic deformation of amorphous layers, as illustrated in Figure 9a. The Cu-CuZr crystal–amorphous nanolaminate with thin amorphous layer thus show uniform plastic deformation. Note that interfaces of crystal–amorphous nanolaminate exhibit no specific crystallographic orientations, activation of STZs in amorphous layer is thus omnidirectional, which can effectively reduce stress concentration. However, the stress in amorphous layers decreases with increasing the thickness of amorphous layers even if crystalline layers experience the same plastic formation. Even though STZs are activated by local stress at interfaces, these STZs slowly propagate across the entire CuZr layer due to the reduced shear stress on the plane non-parallel to the amorphous layers. When the applied stress consciously increases, shear bands form by gathering STZs in the amorphous layers and propagate towards the adjacent interfaces, consequently triggering plastic deformation in the crystalline layers by slip bands, as illustrated in Figure 9b.

### 3.2. Core–Shell Nanostructures

Crystal–amorphous nanocomposite with amorphous GBs has been proven to exhibit both high strength and excellent plastic deformation stability. Take the Ni(75 at.%)-SiOC nanocomposite as an example [53]; in situ micropillar compression tests in a scanning electron microscope (SEM) (Figure 10) demonstrated that the dual-phase core–shell nanostructure exhibited high flow strengths of 2.5 GPa at RT, about 1.6 GPa at 400 °C and an uniform plastic strain of 35% without shear instability. The contribution of amorphous GBs to plastic deformation can be summarized as follows. Firstly, amorphous GBs inhibit grain coarsening during deformation and thus prevent strain localization and localized softening in the shear path, as amorphous GBs can stabilize grain size by decreasing the driving force for GB migration and isolating nanograins from coalescence. Secondly, amorphous GBs impede the motion of dislocations and shear bands, leading to dislocation accumulation at CAIs. Due to low migration and formation energies for defects at CAIs, these accumulated dislocations can be smeared away along CAIs.

For crystalline nanograins, plastic deformation is mainly carried over by dislocations that nucleated at GBs and glide across the entire grain. Corresponding to the fine grain size, dislocations intersect rarely. For fcc metals, high flow strength and fine grain size promote Shockley partial dislocations; consequently, stacking faults are often observed in nanograins. Regardless of the types of dislocations, these dislocations nucleate at GBs and are blocked by GBs. Corresponding to the core–shell nanostructure, dislocations hardly pile up on the same glide plane, and are instead discretely distributed along the CAIs, as illustrated in Figure 11a. Due to low migration and formation energies for defects at CAIs, these blocked dislocations can rearrange easily along CAIs to reduce local stress–strain concentration, thus preventing localized shear associated with planar slips. Furthermore, like the crystal–amorphous nanolaminates, the accumulated dislocations on the two CAIs of the amorphous GB enable the formation and propagation of STZs in the amorphous GBs, achieving plastic co-deformation between amorphous GBs and nanograins. In addition, the random misorientation of NC metals in core–shell crystal–amorphous nanocomposite also contributes to uniform plastic deformation. Compared to crystalline–amorphous nanolaminates synthesized by sputtering, the two neighbouring nanograins often hold well-defined orientation, such as the cube-on-cube orientation relationship or small angle tilt or small angle twist orientation relations (ORs). The good slip continuity across the amorphous layer between two grains (Figure 11b) facilitates deformation compatibility but may lead to crystallographic shear instability, as observed in crystalline–amorphous nanolaminates (Figure 11d,e) [70]. In contrast, in two neighbouring grains with a generally larger angle tilt or twist and tilt–twist coupling ORs, the poor slip continuity (Figure 11c) imparts a strong barrier on slip transmutation across amorphous grain boundaries, preventing crystallographic shear instability. The improved strength by amorphous GBs can be attributed to two aspects: (1) higher strength of amorphous structure than NC metal; (2) reduced GB energy, which may make it more difficult for dislocation nucleation.

### 3.3. Dual-Phase Nanocomposites

For amorphous matrix dual-phase nanocomposites, the nature of plasticity improvement is that the nanocrystal phases can effectively impede the rapid propagation of shear bands while promoting nucleation of multiple fine shear bands or even suppressing the nucleation of shear bands in amorphous matrix. Moreover, ductile nanocrystal phase can co-deform with amorphous matrix, contributing to uniform plastic deformation. For example, Fe(34 at.%)-SiOC amorphous ceramic nanocomposites with self-clustered Fe-rich nanocrystals show a large uniform compressive strain of 55% without plastic flow instability and cracking (Figure 12a,b) [63]. TEM characterization after compression revealed that the spatially distributed Fe nanocrystals plastically co-deform with amorphous SiOC matrix and discretize STZs in amorphous ceramics, thus preventing the shear-banding instability and significantly enhancing compressive plasticity. As shown in Figure 12c,d, Fe nanocrystals exhibit irregular shape and some of them were compressed into stipes in the highly strained region, which provides direct evidence for occurrence of plastic deformation. In addition, no crack or interface decohesion existed in the severely strained regions, indicating the good compatibility of plastic deformation between Fe nanocrystals and the amorphous SiOC matrix. Note that the amorphous-based crystal–amorphous nanocomposites usually show decreased strength with increasing the volume fraction of ductile crystal phase since the NC metals are normally softer than amorphous solid. When nanocrystals are strong or rigid, for example, oxides nanocrystals in bulk amorphous Al_2_O_3_-ZrO_2_-Y_2_O_3_, an unusual behavior of brittle–plastic–brittle transition with increasing crystallinity in amorphous matrix has been reported [106]. Xu et al. [106] discovered that bulk amorphous oxides comprising 3.65 mol% nanocrystallite exhibit a significant plastic strain of 10.5% under compression compared with no plastic deformation occurring in fully amorphous sample at 500 °C, while the plasticity degrades as the crystallinity increases to 24.8 mol%. The plastic deformation was mainly accommodated by shear bands and further densification. The formation of shear bands and the brittle–plastic–brittle transition were explained in terms of free volume theory [127].

For metal matrix dual-phase nanocomposites, amorphous nanoparticles were proved to play an important role in enhancing both strength and plasticity of NC metals. Firstly, the amorphous nanoparticles act as strengthening second phase to impede dislocation motion, pin GBs and prevent grain coarsening, contributing to high strength. Secondly, the nanosized amorphous phase can co-deform with NC metals by STZs or viscoplastic flow to accommodate the shape change in NC metals, achieving superb plastic flow stability. Such a deformation mechanism was demonstrated in 600 °C annealed Ni(75 at.%)-SiOC nanocomposite, which exhibited a characteristic microstructure of 9 nm amorphous SiOC nanoparticles embedded in NC Ni matrix [60]. In situ SEM micropillar compression tests at deformation temperatures ranging from RT to 400 °C revealed that nanograined Ni composite containing amorphous ceramic nanoparticles exhibits high strength of 3 GPa at RT and 2 GPa at 400 °C with compressive strain of up to 50% (Figure 13). The co-deformation mechanism was verified by post mortem TEM characterization. Figure 14a shows a representative TEM bright field image of micropillar after compression. Clearly, both Ni nanograins and amorphous ceramic SiOC particles were compressed into stripes perpendicular to the loading direction. It is also noticed that plastic co-deformation ability decreases with increasing grain size. The 800 °C annealed Ni(75 at.%)-SiOC nanocomposite comprising ~220 nm Ni nanograins and ~110 nm SiOC nanoparticles exhibits low flow strength of 1.5 GPa and no strain hardening. TEM characterizations reveal a slight change in the shape of amorphous SiOC particles associated with deformation, and local interface decohesion between Ni and SiOC particles (Figure 14b). Thus, the softening behavior associated with interface decohesion and the strain hardening behavior associated with dislocations interactions in Ni submicron grains could be counteracted, leading to the absence of strain hardening behavior.

Due to the lack of strain hardening ability in amorphous phase, amorphous phase does not exhibit any ductility. Consequently, cracking often occurs in amorphous phase or amorphous–crystal interfaces when amorphous phase or the interfaces are subject to an effective tension loading [128]. As demonstrated in dual-phase structures with either the core–shell microstructure or metal matrix with amorphous nanoparticles or amorphous matrix with nanocrystals, significantly enhanced compression plasticity is commonly achieved while they exhibit low tensile strength and brittle failure associated with cracking in amorphous phase and de-cohesion of CAIs [128].

## 4. Multiscale Modelling of Mechanical Behaviors

Mechanistic models incorporating deformation mechanisms related to microstructure characters are urgently demanded in order to accelerate the design and discovery of nanostructured composites with specific properties. Given the structural characteristics of dual-phase nanostructures, which include a significant volume fraction of boundaries and nanosized crystalline–amorphous constituents, a multiscale perspective of mechanical behavior is crucial, including but not limited to atomic-scale studies using atomistic simulations and the development of models at different length scales.

### 4.1. Atomistic Modelling

Atomistic models offer valuable insight into the deformation mechanisms of defect–interface interactions, including GB sliding/shear/migration [129], slip transmission [130,131], and dislocation nucleation [132,133], thus shedding light on the underlying deformation processes. In amorphous and crystalline NCs, plastic deformation arises from the coupling between two distinct deformation defects across the CAIs. Dislocations carry the plasticity in the crystalline phase, while the STZ, which is a local shear among a small cluster of atoms, acts as the plasticity carrier in the amorphous phase. Consequently, CAIs exhibit unique inelastic shear transfer properties that are more diffuse, effectively delaying decohesion and fracture initiation near interfaces [56,134]. Figure 15a,b demonstrate the plasticity transfer across the Cu/amorphous-CuNb interfaces simulated by molecular dynamics (MD) simulations [74]. During compression, numerous dislocations nucleate from CAIs, propagate within the Cu layer, and then deposit on the two interfaces that confine the layer. Shear deformation in amorphous CuNb reduces the dislocation content at CAIs, contrasting with the reduction in dislocations at the interface via slip transmission in crystalline metallic multilayers [118]. Figure 15c,d also depict the evolution of atomic structures in the amorphous CuNb layer before and after compressive loading. Deformation distorts the sharp CAIs, indicated by the black dotted lines representing the position of the original layer interfaces and the red dotted lines indicating the location of layer interfaces after 8% compression. More importantly, the plastic deformation induces prominent local shears, evidenced by high-density shear-deformed regions (denoted by three ellipses) across the amorphous CuNb layer. The CAIs can act as strong barriers and traps for impeding and capturing dislocations, thus strengthening the NCs and promoting plastic co-deformation between the crystal and amorphous phases. The accumulated dislocations are distributed along the CAIs, preventing the formation of localized shear bands across multiple nanograins [135,136,137]. The insights gained from atomistic modelling can serve as a guide for the development of mesoscopic and macroscopic continuum models.

### 4.2. Microscale Models

Microscale models with the presumed deformation mechanisms and corresponding physical parameters and functions are able to capture the influence of microstructure characteristics on mechanical responses of NCs [138,139,140,141]. Figure 16 illustrates a quantized crystal plasticity model that was developed to study the distinctive mechanical behaviors of nanocrystalline metals resulting from grain size reduction [142,143]. This model was motivated by molecular dynamics simulations of dislocation loop propagation across nanograins, which showed that the grain-averaged plastic strain jumps by discrete amounts. The activation of the plastic units was based on probability distribution functions related to dislocation nucleation at grain boundaries. The model was used to capture several unique mechanical responses of dislocation plasticity in nanocrystalline metals, including enhanced flow stress, extended plastic transition strain, and a different evolution of intergranular stress with plastic deformation than in coarse-grained counterparts [138,142,143]. Furthermore, a crystal plasticity model for nanolaminates was developed to understand their buckling behavior, as shown in Figure 17 [144]. The model was informed by atomic-level studies of deformation mechanisms [145] and experimental observations [146]. It employed the confined layer slip mechanism, where dislocations propagated within the layer and were deposited at the interfaces. Plastic deformation was assumed to be uniform through the layer thickness, while the variation of elastic deformation through the thickness was captured by dividing each lamella into three or more elements. The model was used to investigate the effect of layer thickness on the buckling behavior of an Al-Al_2_Cu eutectic alloy. The simulated stress–strain responses indicated that the buckling strength and critical strain decreased with increasing layer thickness.

In contrast to the deformation of crystalline phases, amorphous materials can be modelled using isotropic shear transformation. To capture the diverse deformation behaviors of amorphous metals, a mesoscale STZ dynamics modelling framework has been developed [40,41]. This model considered the STZ as the fundamental unit of deformation and used finite element analysis and a kinetic Monte Carlo algorithm to coarse-grain an amorphous collection of atoms into an ensemble of STZs on a mesh. This modelling technique was capable of simulating glass processing and deformation on time and length scales beyond those typically attainable by atomistic modelling. Moreover, it predicted the influence of nanoscale structural heterogeneity on the deformation behavior of amorphous alloys [147,148]. Micromechanical models generally fit empirical functions for fundamental physical quantities and functions to describe constitutive relations of defects, such as the activation energy of STZ, dislocation nucleation rate, and dislocation mobility. However, bridging atomic-level deformation mechanisms and corresponding physical quantities to microscale mathematical and mechanistic descriptions is necessary.

The application of microscale models to predict macroscale behaviors is crucial for the use of large-scale nanostructured materials. While analytical micromechanics methods with reduced orders can provide high efficiency in bridging the gap and homogenizing the heterogeneous material behaviors, they generally lose accuracy or require extensive model calibration when irregular complex morphologies, nonlinear history-dependent properties, or large deformation are present. Consequently, the advancement of machine learning models, such as artificial neural networks and deep neural networks, has emerged as a promising alternative to train surrogate models. As illustrated in Figure 18, the multiscale modeling process can be approximated using a general function *f* to describe the change in the state at a material point with I as the input from the macroscale model and O as the output from the model. The basic idea of using machine learning models to accelerate multiscale modeling is to replace the micromechanics model with a surrogate model constructed from data. The surrogate model can be used to approximate the mechanical behavior of nanocomposites, which is based on the data computed from a series of microscale analyses with various microstructural features. Previous work has demonstrated the feasibility of using neural networks to model constitutive materials, such as hyper-elasticity [149], viscoelasticity [150], elasto-plasticity [151,152], and crystal plasticity [153,154]. These advanced machine learning models are capable of learning complex relationships, thereby eliminating the need for extra calibration of the constitutive laws and micro-mechanical assumption of homogenization. These intelligent modelling methods provide a practical approach to obtain high-fidelity constitutive relations in nanocomposites, where the nanoscale characteristic dimension, non-equilibrium interfacial structure, and complex morphology impart new material laws that require interface-related behaviors to be considered.

## 5. Perspectives

The mechanical behaviors and properties of the crystalline–amorphous dual-phase nanostructures could be optimized by tailoring the morphology, characteristic dimension, and misorientation of nanosized constituents that determine deformation accommodation mechanisms in adjacent constituents and the characteristic size, structure and properties of amorphous phase boundaries that determine the deformation mechanisms within/at/across boundaries. Accelerating the design and discovery of such nanostructured composites with desired properties is thus urgently demanding mechanistic models that can capture microstructure-dominated deformation mechanics and predict the mechanical behaviors and properties. Multiscale material modelling has been extensively employed to understand the mechanical behaviors of nanostructured materials. Atomistic modelling and microscopies at the atomic level can elucidate deformation processes occurring within/at/across boundaries but cannot predict macroscopic properties. Micro-scale mechanistic models have been developed for some characteristic nanostructures (such as nanolaminates) to predict the microstructure-mechanical property relations, but a general approach for the development of microscale models is still missing. More importantly, atomic-level studies and micro-scale mechanistic models cannot directly predict the mechanical response of NCs. This calls for a breakthrough in materials modelling to bridge the length-scale gap from atomic-scale to macro-scale. Machine learning techniques can offer the potential to uncover underlying physics quantities and functions associated with deformation events across different length-scale models and accelerate the development of these constitutive laws.

## Figures and Tables

**Figure 1 materials-16-02874-f001:**
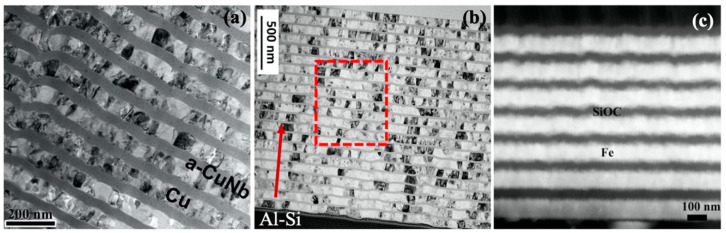
Transmission electron microscope (TEM) images of crystalline–amorphous multilayers synthesized by physical vapor deposition techniques: (**a**) crystalline Cu layers and amorphous CuNb (metallic glass) layers [74], (**b**) crystalline Al layers and amorphous Si layers [71], and (**c**) crystalline Fe layers and amorphous SiOC layers [67].

**Figure 2 materials-16-02874-f002:**
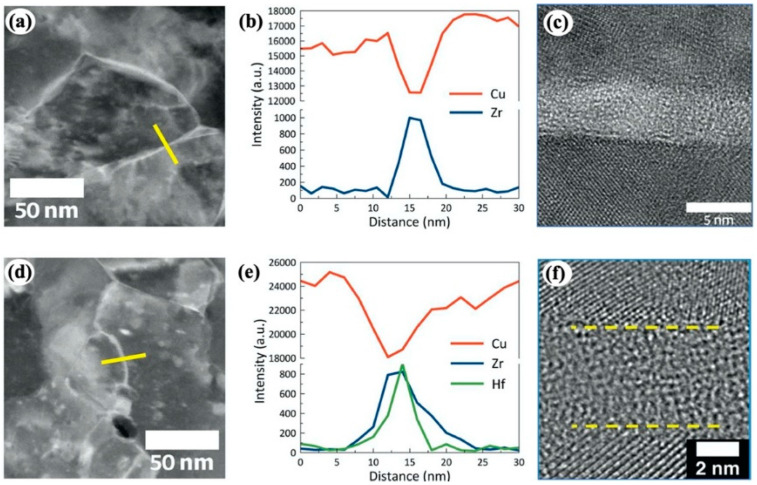
High-angle annular dark-field scanning TEM (STEM) images and STEM-EDS line scans of GB regions for (**a**,**b**) Cu-Zr alloy and (**d**,**e**) Cu-Zr-Hf alloy. Strong segregation of the dopants occurs at GB region. (**c**,**f**) HRTEM images of GBs in Cu-Zr alloy and Cu-Zr-Hf alloy [86]. The bold yellow line in (**a**) and (**d**) indicates the scanning position. The yellow dashed lines in (**f**) show the crystalline-amorphous interfaces.

**Figure 3 materials-16-02874-f003:**
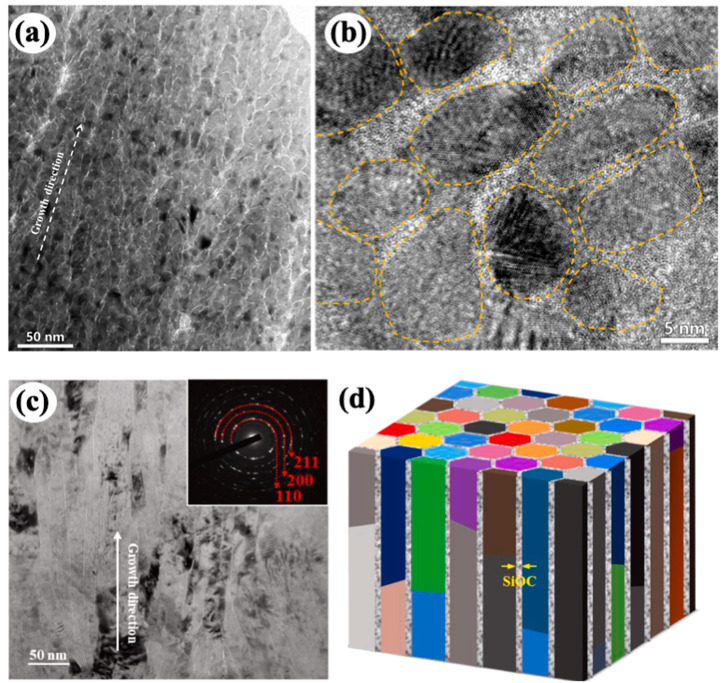
(**a**) TEM image of the as-deposited Ni(75 at.%)-SiOC core–shell nanocomposite and (**b**) HRTEM image showing nanocrystalline Ni and amorphous SiOC GBs [53]. (**c**) TEM image of the as-deposited Fe(80 at.%)-SiOC core–shell nanocolumnars and (**d**) the schematic diagram of the core–shell nanocolumnar structures [72].

**Figure 4 materials-16-02874-f004:**
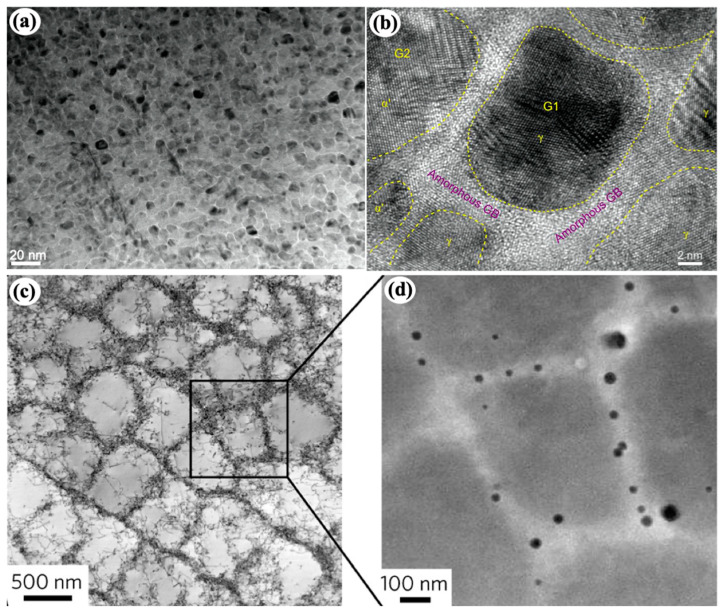
(**a**) A TEM image of laser-processed austenitic Hadfield manganese steel, showing crystalline–amorphous core–shell nanostructure. (**b**) An HRTEM image showing nanocrystals and amorphous GBs [101]. (**c**) A TEM image of solidification cells in austenitic 316 L stainless steels fabricated by L-PBF technique. (**d**) HAADF-STEM image of the solidification cells showing nanoparticles segregated to the cell walls [100,102].

**Figure 5 materials-16-02874-f005:**
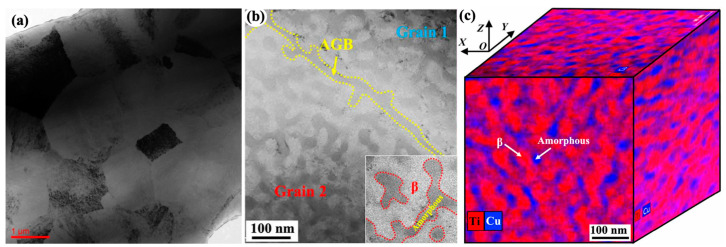
TiZr-based crystalline–amorphous composite [62]: (**a**) A BF-TEM image showing the polycrystalline structure with average grain size of 3 um; (**b**) A TEM image showing amorphous grain boundary, bicontinuous bcc-β (bright) and amorphous (dark) nano-bands; (**c**) A 3D-networked-bicontinuous nanoarchitecture of a grain according to the Ti and Cu distribution maps.

**Figure 6 materials-16-02874-f006:**
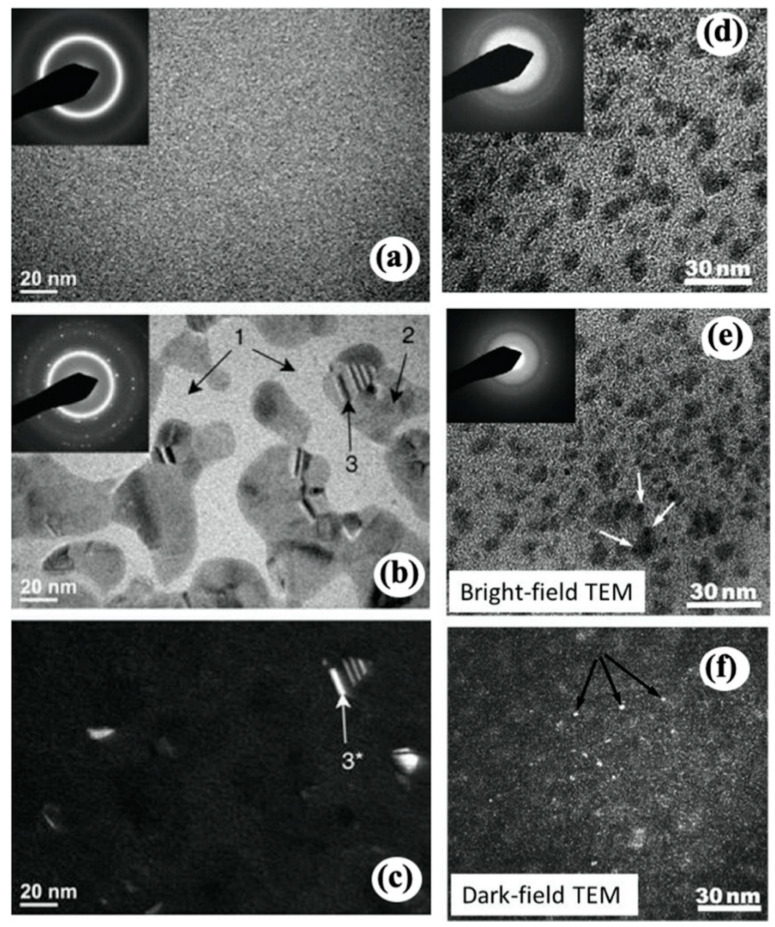
TEM images and corresponding diffraction patterns of (**a**) as-spun Ni_52.5_Nb_10_Zr_15_Ti_15_Pt_7.5_, (**b**) ion-irradiated specimen, and (**c**) the corresponding dark-field TEM image [108]. (**d**) Bright-field TEM image of Cu_50_Zr_45_Ti_5_ metallic glass irradiated by 2 keV Ar ions without cooling and the corresponding diffraction pattern. (**e**) Bright-field TEM image of the specimen in (**d**) after exposure to high-fluence electron beam and the corresponding diffraction pattern [109]. Arrows indicate radiation-induced nanocrystals. (**f**) Dark-field TEM image of the specimen shown in (**d**).

**Figure 7 materials-16-02874-f007:**
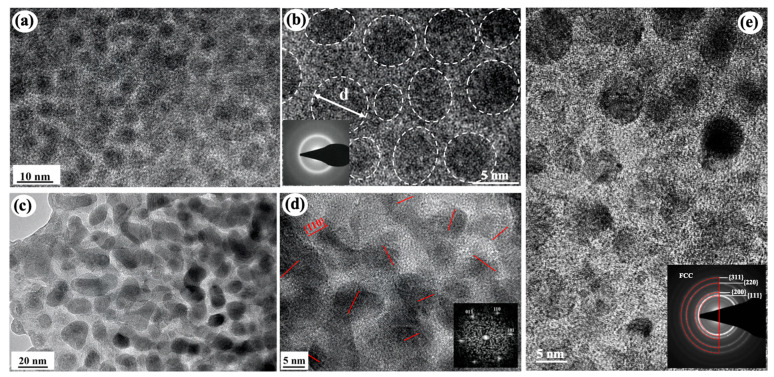
(**a**,**b**) Cross-sectional HRTEM images of the as-deposited Fe(34 at.%)-SiOC dual amorphous composite. The dashed circles indicate Fe-rich amorphous nanoclusters with the diameter of d [63]. (**c**,**d**) Cross-sectional HRTEM images of the 800 °C annealed Fe(34 at.%)-SiOC dual-phase nanocomposite with Fe nanocrystals in amorphous SiOC matrix. The red short lines indicate Fe nanocrystals. (**e**) Cross-sectional HRTEM image of the as-deposited Cu(22 at.%)-SiOC dual-phase composite with Cu nanocrystals in amorphous SiOC matrix [111].

**Figure 8 materials-16-02874-f008:**
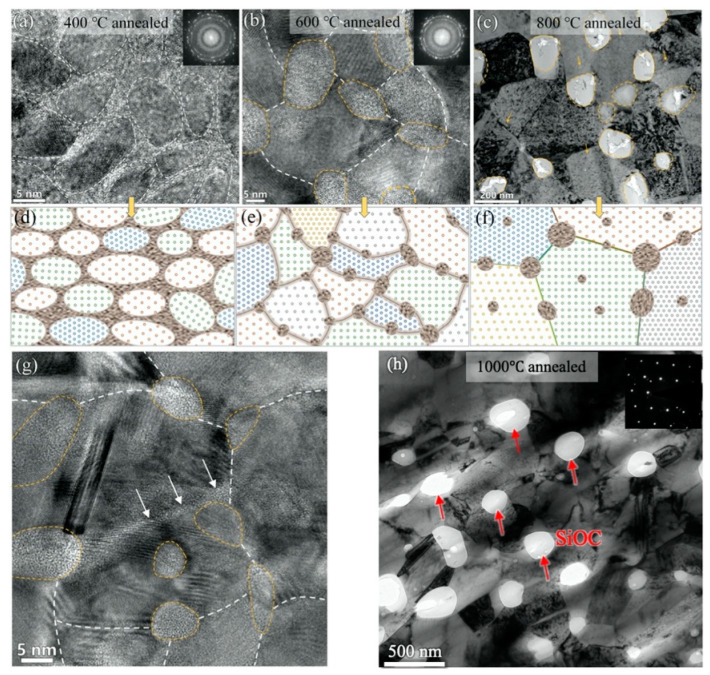
TEM images of Ni(75 at.%)-SiOC nanocomposites after the as-deposited sample was annealed at (**a**) 400 °C, (**b**) 600 °C and (**c**) 800 °C for 1 h [60]. (**d**–**f**) Schematics of characteristic microstructures corresponding to (**a**–**c**). (**g**) A HRTEM image of the 600 °C annealed nanocomposite showing amorphous GBs. (**h**) A TEM image of 1000 °C annealed Ni(75 at.%)-SiOC nanocomposite. The white arrows indicate amorphous Gbs, and the red arrows indicate amorphous particles.

**Figure 9 materials-16-02874-f009:**
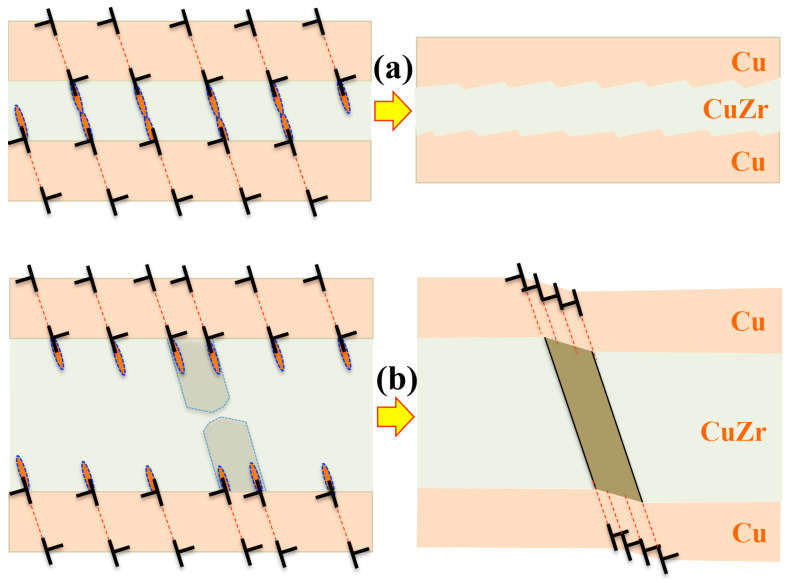
Schematics of plastic deformation modes with respect to the amorphous layer thickness [70]. (**a**) Plastic co-deformation in the thin amorphous layers, showing dislocation slips in the crystal Cu layers and the formation and reaction of STZs in the amorphous CuZr layers. The red ellipses represent STZs. (**b**) Shear banding in the thick CuZr layers where shear bands propagate into the Cu layers via the formation of slip bands on {111} planes.

**Figure 10 materials-16-02874-f010:**
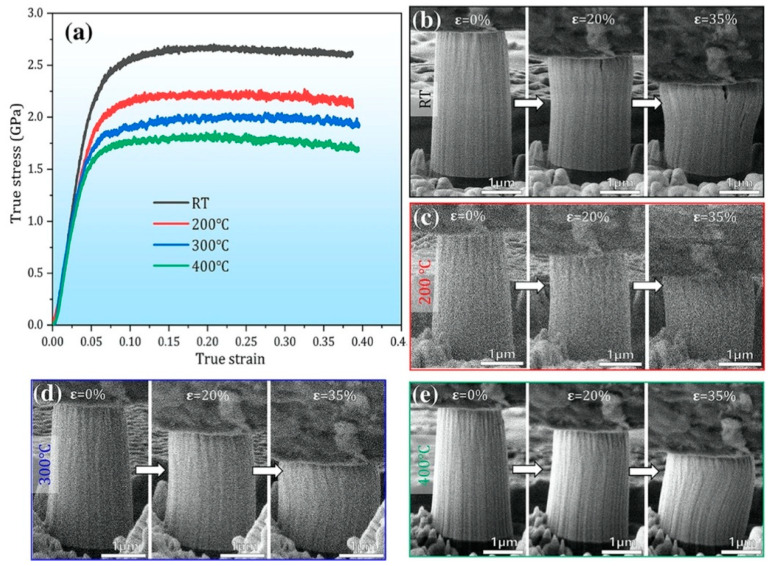
(**a**) The true stress–strain curves of as-deposited Ni(75 at.%)-SiOC nanocomposite at deformation temperatures of RT, 200 °C, 300 °C, and 400 °C [53]. (**b**–**e**) SEM snapshots of the micropillars at different strains.

**Figure 11 materials-16-02874-f011:**
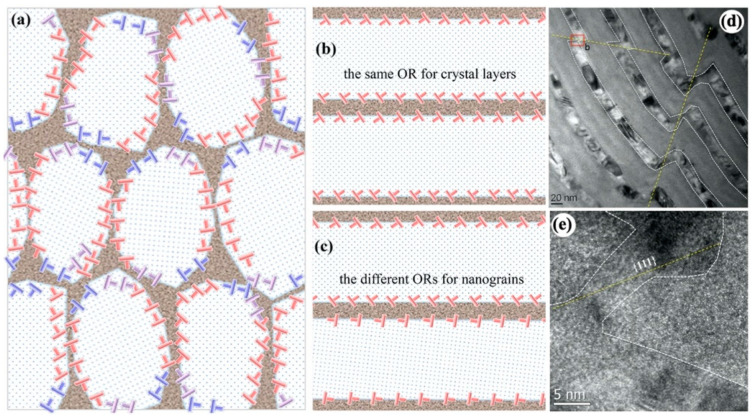
(**a**) A schematic of deformation mechanisms for core–shell nanostructures. Dislocations carried over plastic deformation in nanograins and are blocked at CAIs [53]. (**b**) Good slip continuity for two neighbouring nanograins. (**c**) Poor slip continuity for two neighbouring nanograins. (**d**,**e**) TEM images showing shear banding in Cu/CuZr nanolaminate due to slip continuity in Cu layers [70].

**Figure 12 materials-16-02874-f012:**
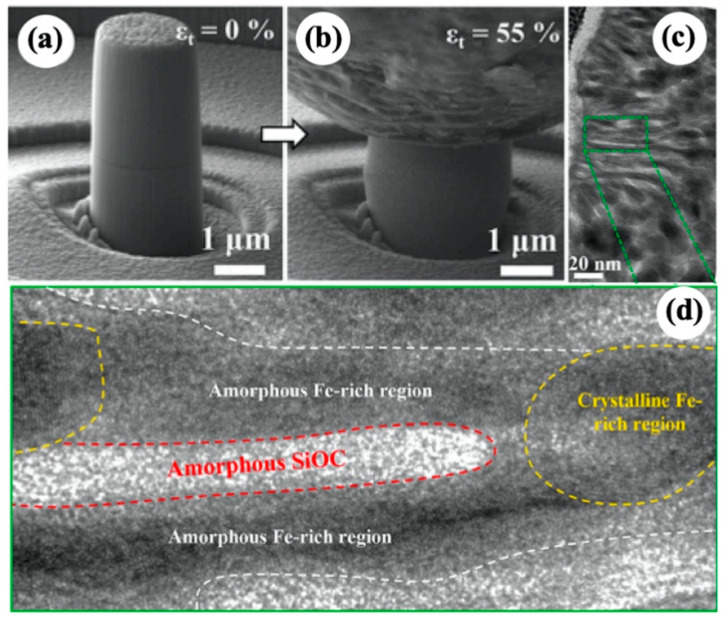
SEM images of Fe(34 at.%)-SiOC micropillar (**a**) before compression and (**b**) after compression test. (**c**,**d**) Cross-sectional TEM images showing co-deformation between Fe nanocrystals and amorphous SiOC matrix [63].

**Figure 13 materials-16-02874-f013:**
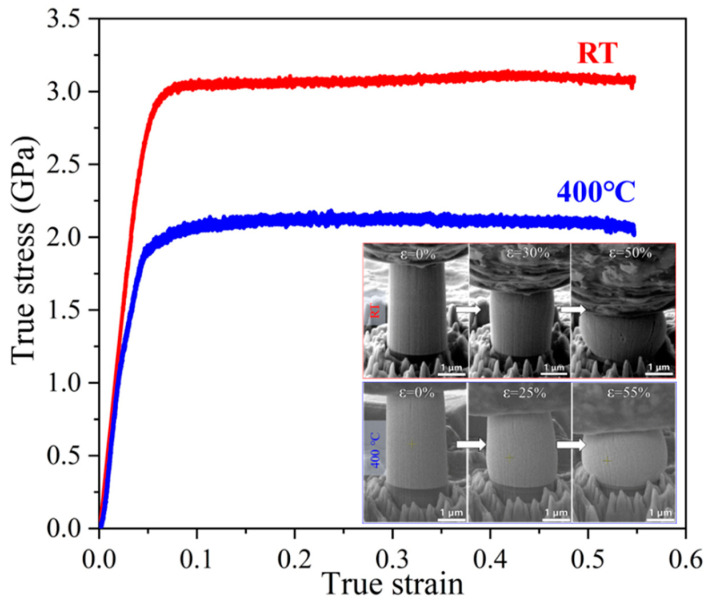
In situ micropillar compression tests of 600 °C annealed Ni(75 at.%)-SiOC nanocomposite at RT and 400 °C [60]. The insets show the corresponding SEM snapshot image micropillars at different strains.

**Figure 14 materials-16-02874-f014:**
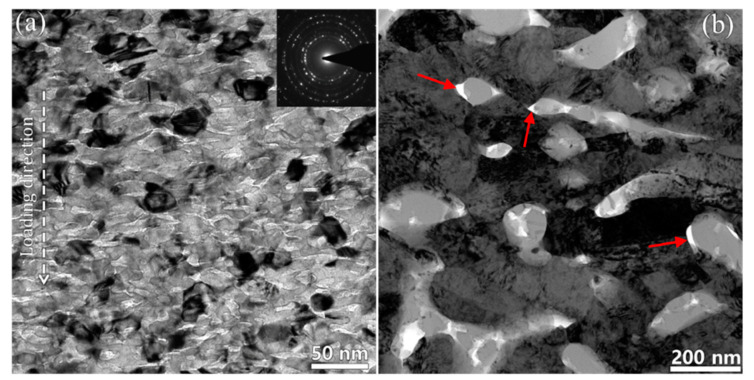
Post mortem TEM characterization of (**a**) 600 °C annealed and (**b**) 800 °C annealed Ni(75 at.%)-SiOC nanocomposites after compression tests [60]. The red arrows in (**b**) indicate interface decohesion between crystal Ni and amorphous SiOC ceramic.

**Figure 15 materials-16-02874-f015:**
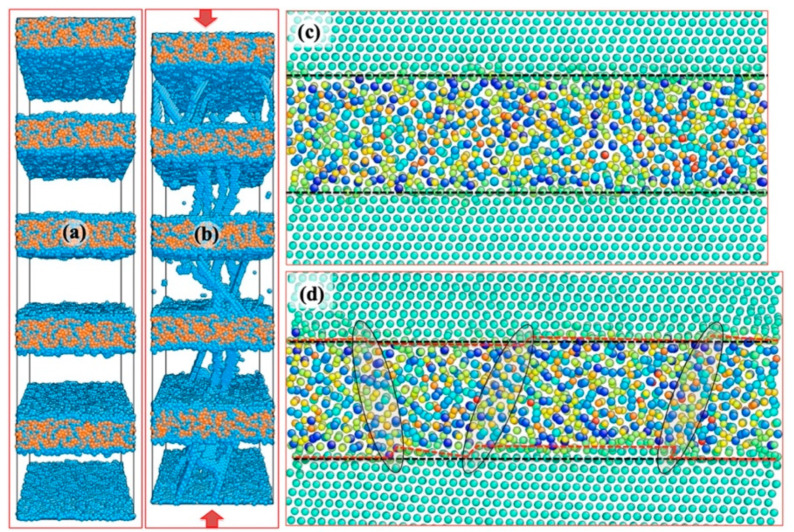
Molecular dynamics simulations of co-deformation in Cu/amorphous CuNb multilayers. (**a**,**b**) The deformed multilayers containing glide dislocations in Cu layers under uniaxial compression. The atomic structures around one amorphous CuNb layer before (**c**) and after (**d**) 8% of compressive strain. The curved CAIs are denoted by the red dotted lines, and local shears of atomic clusters are highlighted by three ellipses [74].

**Figure 16 materials-16-02874-f016:**
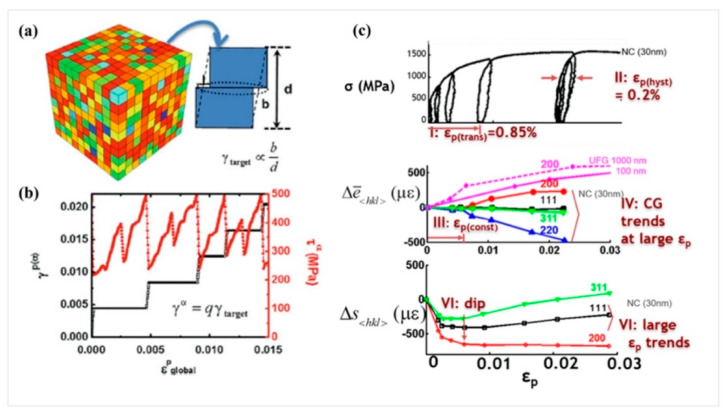
(**a**,**b**) The microscale quantized crystal plasticity model for nanocrystalline metals. (**c**) The modelled stress–strain response along with the stress redistribution in nanocrystalline Ni measured via in situ X-ray diffraction tests [138].

**Figure 17 materials-16-02874-f017:**
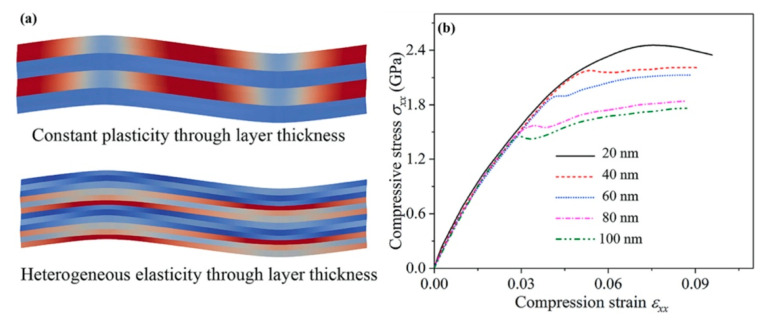
(**a**) A confined layer slip-based crystal plasticity model for predicting the bulking behavior of nanolaminates. (**b**) Effects of the layer thickness on stress–strain curves under parallel compression, and a fine size has good plastic co-deformation and high resistance to compression bulking [144].

**Figure 18 materials-16-02874-f018:**
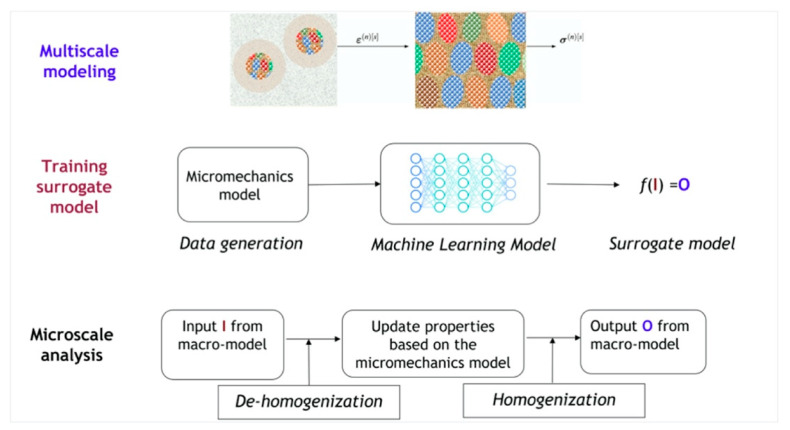
General framework of data-driven multiscale modeling that bridges micro- and macro-scales. Training machine learning surrogate model based on data generated by micromechanics models for prediction of macroscale mechanical behavior of nanocomposite with various nano/micro-structural features.

## Data Availability

No new data were created.

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
