# Peer review of "Crystalline–Amorphous Nanostructures: Microstructure, Property and Modelling"

_materials, 2023, doi:10.3390/ma16072874_

Round 1
Reviewer 1 Report
This manuscript is topic of review on Crystalline-Amorphous Nanostructures: Microstructure, Property and Modelling. I would like to say that the study is only acceptable after major revision, because manuscript needs to be improved. recommendations are attached as file
1.

Author Response
This manuscript is topic of review on Crystalline-Amorphous Nanostructures: Microstructure, Property and Modelling. I would like to say that the study is only acceptable after major revision,
because manuscript needs to be improved. recommendations are listed below.
- A comprehensive explanation of the research was not made in the abstract section. Especially the usage areas of amorphous materials and their importance in new technologies should be emphasized more clearly.
Reply: The focus of this review paper is crystalline or amorphous based dual-phase nanocomposites. In the abstract, we concisely elucidated the motivation for designing such a nanocomposite and more comprehensive explanation was presented in the introduction part. The potential usage area of crystal-amorphous dual-phase nanocomposites here is for advanced nuclear reactors, and that is also why we focused on their thermo-mechanical and irradiation resistance properties. We clarified this in Abstract.
- In the introduction section, examples of fillers or metal materials used in the synthesis of
amorphous nanomaterials for the development of high strength and weak deformation properties and research findings in the literature should be added in summary. These details are quite clear in the manuscript, but there is not enough explanation in the introduction part. At the same time, details such as examples of materials used to improve thermal properties, or details of new technologies applied to improve deformation ability, and research findings should be added in addition to how many percent it improves.
Reply: Thanks for your suggestions. In the introduction section, we aim to elucidate the background and motivation for designing the crystal-amorphous nanocomposite, and our focus is the general background and mechanism. The details and examples were presented in the manuscript associated with the characteristic microstructure. Note that it is difficult to quantify the improvements of thermo-mechanical properties due to the wide variation of characteristic microstructures and compositions of crystal-amorphous nanocomposites. Qualitatively comparing with the monolithic material system (nanocrystal metals or amorphous solids) should be enough. For example, the amorphous solids usually exhibit “brittle behaviour” associated with formation of shear banding and cracking, while crystal-amorphous nanocomposites show high plastic flow stability.
- At the same time, amorphous nanocomposites were synthesized in the following article. It
will be useful to use in this study.
Improvement of Synthesis and Dielectric Properties of Polyurethane/Mt-QASs+ (Novel Synthesis) G Baysal, H Aydın, H HoÅŸgören, S Uzan, H Karaer Journal of Polymers and the Environment 24, 139-147.
Reply: We have cited the reference in the revised manuscript.
- Materials that are effective in providing high radiation tolerances of amorphous materials need further specification.
Reply: Compared to crystalline materials, the amorphous structure in amorphous solids that intrinsically do not contain crystal defects could offer the possibility of eliminating the root cause responsible for radiation damage in crystalline materials. We point this out in the introduction part: “Especially high crystallization temperature amorphous materials could be very promising candidates for high radiation tolerance at high temperatures since they do not contain conventional crystal defects such as vacancies, interstitials, or dislocations which evolved in crystalline materials under irradiation” (lines 9-12, page 2).
- In the conclusion part, the usage areas that amorphous materials gain advantage and their contributions to new technologies should be explained more effectively. This is also of great
importance in emphasizing the importance of the study.
Reply: The topic of this review paper is Crystalline-Amorphous Nanostructures: Microstructure, Property and Modelling. We care more about the characteristic microstructure, thermo-mechanical behaviour, irradiation resistance, deformation mechanism and multiscale modelling. Such crystal-amorphous nanocomposites are potential candidates for advanced nuclear reactors.
- Concentrations of metallic materials used in the synthesis of amorphous materials are very
important, especially in the synthesis of biomaterials. An endless improvement cannot be made, nor an unlimited usage rate. In similar studies, some limit values should be specified in order not to threaten health.
Reply: In this review paper, our focus is the thermo-mechanical and irradiation resistance properties of crystal-amorphous nanocomposites. The potential application of acting as biomaterials is out of the scope of this review paper.
- In some parts of this review, plagiarism is quite disturbing. Please make improvements.
Reply: Note that most the results/explanation in the manuscript are from our own previous works. We have rewritten these parts in the revised manuscript.
- Minor grammatical errors in English need to be corrected.
Reply: We have corrected these errors and carefully checked the entire manuscript.
Reviewer 2 Report
The reviewer found the idea of the submitted manuscript of the title ‘Crystalline-Amorphous Nanostructures: Microstructure, Property and Modelling’ However, it is needed to be revised according to the following comments with a minor revision before publication.
1. Authors have complied the manuscript in a competent manner. However, I would suggest including a few more studies, for example, the authors include a few more relevant studies for annealed and unannealed samples to support their existing study. As temperate is the very dominant factor for the consideration.
2. I hardly see any mathematical modeling with simulation. This type of study can increase the impact of the paper.
3. ZnO is considered a significant semiconductor oxide, I guess it would be better if authors can include a few studies of ZnO-related reports. Please see the following:
a. Ultraviolet Photodetection Based on High-Performance Co-Plus-Ni Doped ZnO Nanorods Grown by Hydrothermal Method on Transparent Plastic Substrate
b. High-performance flexible ultraviolet photodetectors with Ni/Cu-codoped ZnO nanorods grown on PET substrates
Author Response
The reviewer found the idea of the submitted manuscript of the title ‘Crystalline-Amorphous Nanostructures: Microstructure, Property and Modelling’. However, it is needed to be revised according to the following comments with a minor revision before publication.
- Authors have complied the manuscript in a competent manner. However, I would suggest including a few more studies, for example, the authors include a few more relevant studies for annealed and unannealed samples to support their existing study. As temperate is the very dominant factor for the consideration.
Reply: The mechanical behaviour of crystal-amorphous nanocomposites is highly dependent on their characteristic microstructures such as the feature size and distribution of crystal or amorphous constituent. As presented in our manuscript, the characteristic microstructures include nanolaminates, core-shell microstructures, and crystalline/amorphous-based dual-phase nanocomposites. High temperature annealing is just a strategy to tailor the microstructure and check the structure stability of crystal-amorphous nanocomposites. For example, the as-deposited Ni(75at.%)-SiOC exhibit core-shell structure and maintain stable up to 400 ℃. After 600 ℃ annealing, the core-shell structure transformed into crystalline based dual phase nanocomposite with amorphous nanoparticles embedded in the Ni matrix. The different microstructures corresponding to different mechanical behaviours and deformation mechanisms.
- I hardly see any mathematical modeling with simulation. This type of study can increase the impact of the paper.
Reply: As described in the perspective section, Micro-scale mechanistic models have been developed for some characteristic nanostructures (such as nanolaminates) to predict the microstructure-mechanical properties relations, but a general approach for the development of microscale models is still missing. More importantly, atomic-level studies and micro-scale mechanistic models can not directly predict the mechanical response of NCs. This calls for a breakthrough in materials modelling to bridge the length-scale gap from atomic-scale to macro-scale.
- ZnO is considered a significant semiconductor oxide, I guess it would be better if authors can include a few studies of ZnO-related reports. Please see the following:
- Ultraviolet Photodetection Based on High-Performance Co-Plus-Ni Doped ZnO Nanorods Grown by Hydrothermal Method on Transparent Plastic Substrate
- High-performance flexible ultraviolet photodetectors with Ni/Cu-codoped ZnO nanorods grown on PET substrates
Reply: We have cited above references in the revised manuscript.
Reviewer 3 Report
1. The detailed description of synthesis: which substrates were used, which materials were obtained, what was the stoichiometry of synthesis, and why.
2. Please specify the conditions of synthesis, not only the cooling rate.
3. What was the result of synthesis, lack of chemical compositions of obtained compounds.
Chemical compositions should be presented in the Table.
4. The authors focused only on one research method (TEM). Why did the authors not apply other techniques: XRD, DSC/TG, magnetic and electrical measurements?
5. The presented materials are not appropriately described.
6. The manuscript is prepared chaotically. It should be orderly. Materials should be collected in groups, presented, and described according to their chemical composition
I cannot recommend this article in present form.
Author Response
- The detailed description of synthesis: which substrates were used, which materials were obtained, what was the stoichiometry of synthesis, and why.
Reply: As presented in the manuscript, the crystal-amorphous nanocomposites were classified according to their characteristic microstructures, such as nanolaminates, core-shell microstructures, and crystalline/amorphous-based dual-phase nanocomposites. For the sample synthesis, we focus on the preparation methods and basic thermodynamic mechanisms. Detailed stoichiometry can be found in each related reference. The details of composition and synthesis condition for each nanocomposite are not the subject of this review paper.
- Please specify the conditions of synthesis, not only the cooling rate.
Reply: In this review paper, we focused on the characteristic microstructure, thermo-mechanical and irradiation resistance properties, and multiscale modelling of crystal-amorphous nanocomposites. Take the sputtering deposition as an example, there are many details for the synthesis condition, such as target composition, target type and number (co-sputtering), base and Ar pressure during deposition, input power, deposition time and substrate-target distance. Too many details for the composition and synthesis condition will make the review paper complicated and they are not the subject of this review paper.
- What was the result of synthesis, lack of chemical compositions of obtained compounds.
Chemical compositions should be presented in the Table.
Reply: The topic of this review paper is Crystalline-Amorphous Nanostructures: Microstructure, Property and Modelling. The detailed chemical composition can be found in the related references.
- The authors focused only on one research method (TEM). Why did the authors not apply other techniques: XRD, DSC/TG, magnetic and electrical measurements?
Reply: In this review paper, we focused on the characteristic microstructure, thermo-mechanical and irradiation resistance properties, and multiscale modelling of crystal-amorphous nanocomposites. TEM is enough to reveal the feature size, phase distribution, composition and deformation mechanism of crystal-amorphous nanocomposites.
- The presented materials are not appropriately described.
Reply: We classified the crystal-amorphous nanocomposites based on their characteristic microstructure, i.e., nanolaminates, core-shell microstructures, and crystalline/amorphous-based dual-phase nanocomposites. Based on each characteristic microstructure, we give examples to reveal the thermo-mechanical and irradiation resistance properties and the corresponding deformation mechanisms.
- The manuscript is prepared chaotically. It should be orderly. Materials should be collected in groups, presented, and described according to their chemical composition.
Reply: The topic of this review paper is Crystalline-Amorphous Nanostructures: Microstructure, Property and Modelling. We classified the crystal-amorphous nanocomposites based on their characteristic microstructure, i.e., nanolaminates, core-shell microstructures, and crystalline/amorphous-based dual-phase nanocomposites. Based on each characteristic microstructure, we give examples to reveal the thermo-mechanical and irradiation resistance properties and the corresponding deformation mechanisms.
Round 2
Reviewer 1 Report
Dear Authors, I am pleased to inform accept of revised manuscript.
Reviewer 3 Report
After correction and explanation, I can recommend this manuscript for publication.